# Development and Validation of a New Questionnaire to Measure Knowledge Level of Street Food Hawkers to Support the Single-Use Plastics Reduction Program in Kelantan, Malaysia

Nur Baizura Aini Abdullah, Nor Azwany Yaacob and Ahmad Filza Ismail *

Department of Community Medicine, School of Medical Sciences, Health Campus, Universiti Sains Malaysia, Kubang Kerian 16150, Kelantan, Malaysia; drbaizura@student.usm.my (N.B.A.A.); azwany@usm.my (N.A.Y.)
* Correspondence: afilza@usm.my

**Abstract:** Food hawkers' knowledge about single-use plastic food containers must be assessed using a valid and reliable tool to improve assessment and evidence collection credibility and to promote environmental sustainability practice. This study aims to develop and validate a new questionnaire to assess the knowledge level of street food hawkers to support the single-use plastics reduction program. Seven experts were involved in the questionnaire process. In the validation process, a cross-sectional study employed the purposive sampling of 660 night-market street food hawkers in a north-eastern state in Peninsular Malaysia, utilising a Google Forms questionnaire with 22 self-administered items. The Content Validity Index (CVI) and Face Validity index (FVI) were used for the construct analysis. The dichotomous response scale was analysed using a two-parameter logistic model of item response theory (2-PL IRT), while marginal reliability used to determine the internal consistency. The I-CVI were calculated for all items with the value $\geq 0.83$, except for nine items with I-CVI < 0.83. I-FVI with values of 0.83 or more were acceptable. The 2-PL IRT analyses indicated good psychometric properties considering the discrimination and difficulty index. The marginal reliability value was 0.77. The newly developed questionnaire is a valid and reliable tool to assess the knowledge level of street-food hawkers to support the single-use plastics reduction program.

**Keywords:** single-use plastic food container; street food hawker; plastics; knowledge; content validity; face validity; item response theory

## 1. Introduction

Single-use plastics, also known as disposable plastics are commonly used for packaging and are intended to be used once and then discarded or recycled [1]. Plastic is commonly utilised for goods such as plastic bags, straws, disposable bottles, food containers, and others, due to its desirable aspects, such as being lightweight, strong, inexpensive, and versatile [2]. Packaging waste is among the most pressing global environmental issues [3] because packaging accounts for extensive plastic use. Most food packaging is designed for single-use and is not recycled. After plastic packaging materials have served their purpose, most are disposed of as waste [4].

The production, consumption, and management of single-use plastic food containers poses problems to human health and the environment. Single-use plastics pose even more significant hazards since they degrade into smaller particles (microplastics) and eventually infiltrate the water and food chain [5]. Plastic waste clogs sewers, causing floods and disease, and it may also find its way into the food chain if consumed by livestock [1]. Microplastics can enter the human body via the digestive tract through food consumption; microplastics have been found in human faeces [6]. In addition, microplastics can leak from plastic bottle caps [7] and plastic teabags [8]. Plastics might release several chemicals,

such as bisphenol A (BPA), phthalate, and styrene that accumulate in the human body and impair organ function [9]. Phthalate is used for plastic food wrapping. It is an "endocrine disruptor" that interferes with hormone production [10].

The world has most recently produced 275 million tonnes of plastic waste, exceeding the previous record of 270 million tonnes [11]. In 2019, Malaysians generated plastic waste at a rate of 16.78 kg/person/year and higher, compared to countries such as China, Indonesia, Philippines, Thailand and Vietnam [12]. In Malaysia, the consumption of single-use plastics and packaging for consumer goods has gradually increased over the past few decades.

Overcoming the elimination of single-use plastics is one of the biggest ecological problems in recent times. While there is a sense of urgency and increased attention on plastics as an environmental issue by industry and governments, there is a gap concerning public knowledge data, including food hawkers' data, for supporting the plastic reduction program. Understanding the broader background regarding food hawkers' knowledge concerning environmental issues is also essential to reducing excessive plastic use. Environmental knowledge can be defined as the ability to identify several symbols, concepts, and behavioural patterns that are related to environmental protection [13]. Some studies have found that people with more knowledge of environmental issues are likely to demonstrate pro-environmental behaviour [14]. A transtheoretical model explained the stages that are involved for behavioural change to occur. It describes six stages of behavioural readiness: pre-contemplation, contemplation, preparation, action and maintenance [15]. Knowledge may affect the readiness of the individual to support pro-environmental behaviour, and possessing correct knowledge has been shown to predict pro-environmental behaviour [16].

Past studies mostly stressed environmental concerns, such as ocean pollution, waste generation, and isolated health effects due to plastic use. Research that was carried out on food hawkers emphasised food safety and food hygiene, with very few studies focusing on using single-use plastics as food containers.

One example of a study focusing on single-plastic usage as food containers was a study on single-plastic usage using a questionnaire examined the perception of plastic packaging that is used to pack hot foods among food hawkers at the night market in Malaysia. The questionnaire was adapted from related prior studies with a questionnaire validation process that was carried out by selected panel experts with an acceptable Cronbach alpha between 0.55 and 0.89 [17]. The triggering factor of this study was the awareness of cancer triggers among Malaysians related to the health hazards of plastic leaching chemicals into food. Our study evaluates the general single-use plastic of food containers that hold hot food, desserts, and drinks. We also include the health impacts of plastic use on high-risk populations, such as babies, children, and pregnant women, as well as the fate of plastic, chemical hazards in plastic, plastic's environmental impact, and laws that are related to plastic usage. Another study involved 300 food handlers in Egyptian universities on KAP and focused on a selected plastic-type food contact material. This study utilized a validated survey instrument with a good internal consistency of Cronbach's alpha value of more than 0.7 [18].

Given the lack of questionnaires concerning street food hawkers' knowledge on the single-use plastics reduction programme and the absence of a Malay questionnaire, it is critical to construct a valid and psychometrically sound questionnaire. In addition, previous studies on plastic-use knowledge focused on consumers as the target population, while few studies targeted the food hawkers. Survey-based studies are important tools in social, medical, economic and behavioural research and questionnaires are used to measure a variety of information provided by the participants As a result, in establishing the quality and scientific worth of any survey-based research, the use of a well-designed questionnaire is critical [19]. Designing a questionnaire is crucial in determining only the relevant items that reflect specific constructs to be measured in research, and questionnaire validation is a crucial step to ensure quality responses and results [20].

This study aims to develop and validate a new questionnaire to assess the knowledge of street food hawkers in Kelantan, in order to support the single-use plastic reduction program. This study can help to determine public acceptance and support for the plastics reduction program. Food hawkers will experience changes in their plastic-use system because they use plastics for business and might thus be the ones to initiate the plastic reduction program. It is evident that any improvements in plastic use must be economically and technically feasible and socially acceptable [21]. The questionnaire must also be tailored to the local culture, beliefs, and habits to capture valuable comparative data. Validity and reliability studies are essential to increase questionnaire credibility as a research tool to produce valid data. They help to provide good quality data with high comparability and credibility, allowing generalisation for a broader population.

## 2. Material and Methods

### 2.1. Study Designs and Participant

This study involved the development and validation of a new questionnaire on street food hawkers' knowledge to support the single-use plastics reduction program as summarized in Figure 1.

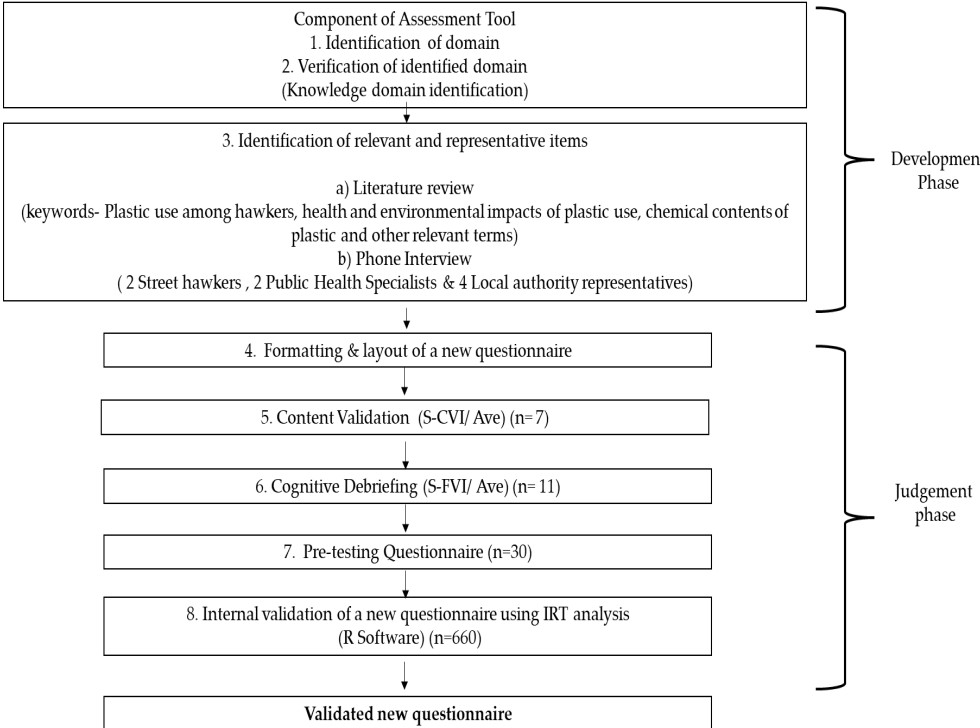

**Figure 1.** The flowchart of development and validation of newly developed questionnaire.

This cross-sectional validation study was conducted in Kota Bharu, a district in Kelantan state which is situated in a north-eastern state in Peninsular Malaysia. Kota Bharu is the capital city of Kelantan. It is highly populated, containing about 32% of the Kelantan population [22]. Development and validation were conducted between June 2020 and February 2021. The questionnaire development involved 7 experts: 3 public health specialists, two local authorities' representatives, a toxicologist, and a lecturer from the linguistics department.

In the validation process, purposive sampling was used to gather a total of 660 respondents of food hawkers from 22 night markets in Kota Bharu. A minimum sample of 500 participants was adequate to conduct an analysis for a dichotomous unidimensional model. An accurate estimate of item parameters can be obtained with 500 respondents and 20 items to be analysed [23]. The inclusion criteria were Malaysian street food hawkers

aged 18 years old and above who were able to read and understand the Malay language and were willing to participate in the study.

*2.2. Questionnaire Development*

A set of new questionnaires was developed through a literature review and in-depth phone interviews with six representatives. The interviews comprised two street hawkers, four Kelantan local authority representatives, and two public health specialists. These interviews helped to explore food hawkers' practices on plastic food container use, their awareness of the plastics reduction program, knowledge of chemicals, and the health and environmental effects of single use plastics. The literature review used keywords concerning food hawkers' knowledge about plastic use, health and environmental impact, chemicals used in plastic food containers, and other relevant terms from journals databases such as EBSCOhost, ScienceDirect, Google Scholar, PubMed, and others. These two methods helped to identify the relevant domain or construct definition and related theory.

The identified domains were validated by assessing them for their appropriateness and applicability to the target population. Several meetings were arranged with an expert panel to obtain domain-specific views, construct conceptualisation, and determine the possible language that respondents might use to describe the domain during verification. The proposed items were developed to represent the construct so that the respondents could easily understand it and answer the research questions. Double-barrelled questions were asked separately to avoid confusion [24] and ensure that the proposed items included a range to verify the questionnaire's difficulty [25]. Final item and domain verification comprised expert validations on the proposed domains and items, including representativeness, clarity, relevance, and distribution.

The questionnaire was then adjusted for layout and formatting to increase presentability. The items were reworded, adjusted to prevent bias, and rephrased to avoid double meanings. A nominal scale was used to measure target knowledge using "yes", "no", and "don't know" options. The Modified Delphi technique was implemented for the questionnaire development process [26]. This technique comprises providing experts with a set of items to rate their importance. Open-ended questions that are selected by the research team are used for the typical Delphi technique. The present technique is widely used to obtain unbiased and systematic expert consensus on a topic. This is the first draft of the newly developed questionnaire.

*2.3. Validation of the Questionnaire*

Validity explains how well the collected data cover the actual area of investigation [27]. This questionnaire underwent a content validation and face validation process in the judgement phase.

2.3.1. Content Validity

Content validation is the process of examining the item contents and whether or not they represent the entire theoretical construct of the designed model [28]. This is measured through the Content Validation Index (CVI). The CVI includes the assessment of relevance and representativeness of each item by the panel of experts.

The content validation form was emailed to seven experts who agreed to join the validation process. The seven experts were 3 public health specialists, 2 representatives from local authorities, a toxicologist, and a lecturer from the linguistics department. The lecturer from the linguistics department was selected to assist in the items' linguistic elements. The experts reviewed the relevance of the proposed items to the measured knowledge domain based on a 4-point scale. A scale of one indicated that the item was not relevant, while four indicated that the item was highly relevant to the measured knowledge domain. The experts were provided with knowledge domain definitions, terminology, and specific instructions to aid them in this process.

Additionally, all the experts were encouraged to provide written comments before providing the score for each item. All suggestions were considered for the improvement of an item's clarity, its relevance and the sentence that was used. The completed content validation forms were emailed to the researchers for improvement. The second draft was created after this process.

### 2.3.2. Face Validity

A face validity test was conducted with selected respondents to determine how easily they understand the proposed items, including their feasibility, formatting style, readability, and lexical clarity. This was to ensure that the respondents interpreted the item as it was intended to be understood by the research team. The respondents were asked to actively voice their thoughts as they attempted to answer the questionnaire draft [29]. The outcome of this process can identify the mistakes that respondents may make in their interpretation of the question or the response options. The process also aimed to identify difficult and confusing items [30].

The raters of face validity include the non-professional person who takes the test and cannot be replaced by the expert, professional, or psychometrician. There were 11 individuals involved in the face validation process: 5 representatives from local authorities and 6 members from a hawker's association. The questionnaire and proper instruction were given through Google Forms. Each item was reviewed on its clarity and comprehension based on a 4-rating scale: 1—the item is not clear and not understandable, to 4—the item is very clear and understandable. This process produced the third draft of the questionnaire.

### 2.3.3. Pretesting Process of Validity

The third draft of the questionnaire was distributed to 30 restaurateurs or food stall workers in Pasir Puteh district in Kelantan. The sample size was 30 according to Roscoe rule of thumb for a simple exploratory study [31]. The questionnaire was prepared in the Malay language in a self-administered Google Form as the data collection was carried out during the COVID-19 pandemic. The respondents were given a QR code in the consent form, and were instructed to scan the barcode to achieve access to the questionnaire.

### 2.3.4. Item Response Theory

A psychometric evaluation is completed to check the internal structure of this questionnaire. The item–item, item construct and construct–construct interrelationship is measured. This is achieved by factor analysis and reliability. In this study, Item Response Theory (IRT) can provide an assessment of quality measures into the difficulty and discriminative ability of the test items that are proposed in this new questionnaire. It also allows analysis of the dichotomous responses assessments which were used in this questionnaire [32]. In this study, the fourth draft of the questionnaire was distributed to 660 night-market street food hawkers at Kota Bharu district, Kelantan state. The questionnaire was distributed through a QR code due to the COVID-19 pandemic.

### 2.4. Statistical Analysis

In the judgement phase, each item underwent an assessment of Content Validation Index (CVI) and Face Validation Index (FVI), and an Item Response Theory (IRT) analysis. The CVI assessed the relevance and representativeness of each item by experts based on a 4-rating scale. The raw panel rating was entered into Microsoft Excel. For objective assessment of the content validity, an item-level content validity index (I-CVI); scale-level content validity index (S-CVI); and S-CVI/Ave based on a formula by Yusof (2019) were used [33]. The relevance rating was recorded as 1 (relevance scale of 3 or 4) or 0 (relevance scale of 1 or 2). Since more than five experts were involved in the evaluation, the lower limit of I-CVI and S-CVI/Ave was more than 0.83 [34]. Items with an I-CVI value of more than 0.83 were retained in this questionnaire.

A face validity index (FVI) is carried out to assess the clarity and understanding rating of the item as assessed by respondents. The items were scaled based on a 4-point Likert scale and entered into Microsoft Excel. The I-FVI and S-FVI/Age were calculated in this process to assess the average of the clarity and comprehension scores across all raters [35]. An FVI value of 0.83 or more was considered good for the online survey by more than 10 raters (11 raters were responded in our study) [36].

An Item Response Theory (IRT) analysis was performed for the internal validity analysis. The Two Parameter Logistic Model (2PL) IRT analysis was used to estimate the difficulty and item discrimination parameter for items with a dichotomous response. The response options were "yes", "no" and "don't know". The response option was coded into a dichotomous outcome of "1" for a correct answer and an incorrect response coded was "0". The IRT analysis was run using a "mirt" package under R software version 1.4.1106. The difficulty and discrimination indices were analysed to decide on the quality of items in the test. A difficulty parameter ability of $-3$ to $+3$ was chosen as this is considered a typical and practical range [32]. Items with a difficulty parameter of <3.0 or >3.0 were discarded. For discrimination parameters, the cut-off point of 0.8 to 2.5 was kept in the analysis as it had a good discrimination parameter [37]. Unidimensionality in IRT models refers to the assumption that the measured construct is unidimensional, that is, that the covariance among the items can be explained by a single underlying dimension. Examination of output from an exploratory factor analysis, including eigenvalues, scree plots, and the magnitude of item loadings on the first factor can help in evaluating this assumption. A second assumption of IRT models is that the items display local independence. This is technically subsumed under the unidimensionality assumption and requires that, given their relationship to the underlying construct being measured, there is no additional systematic covariance among the items [38].

Assessments on model fitness were carried out using the assessment item fit and model fit criteria of the Root-Mean-Square Error of Approximation (RMSEA), Comparative Fit Index (CFI) and Tucker–Lewis Index (TLI). Item fitness was assessed based on a Chi-square goodness-of-fit test if the p-value was $\geq 0.05$. The assessment of goodness-of-fit for two-way margins was carried out to assess the Chi-square of residuals, where a Chi-square of a value of >4 indicates a poor fit on the two way margin [32]. RMSEA is an absolute fit index in that it assesses how far the initial proposed model is from a perfect model. The cut-off point value of RMSEA that was taken for this study was <0.08 which suggests a reasonable model-data fit [39]. A CFI and TLI larger than 0.95 indicate a relatively good model–data fit in general [40]. Model fitness can also be proven through the Akaike Information Criterion (AIC) and Bayesian Information Criterion (BIC) value, where a model with a lower AIC was chosen in the model fitness process. An improvement in the model is shown as a reduction in AIC and BIC values throughout the model review process [41].

The marginal reliability can estimate the average reliability among the respondent's knowledge [42] and shows the overall consistency of the test scores in measuring the observed score of the underlying trait that is generated from the IRT model [43]. The value of marginal reliability of 0.623 as suggested by Dimitrov was used in our study [40].

## 3. Results

### 3.1. Content Validity of the Questionnaire

The first draft of the questionnaire consisted of six domains with 65 items. The factors and the number of items are shown in Table 1.

In this process, the comments indicated that the suggested items were too difficult, too easy, irrelevant, requiring rewording, and having inappropriate double-barrelled questions. Three items concerning the health effects of plastic use and one item from knowledge on plastic categories were removed. The CVI result is shown in Table 1, and four items were removed in this process. The second draft comprised 61 items with six factors that were used for the cognitive debriefing process.

**Table 1.** Content Validity Index (CVI) of Draft 1 questionnaire (n = 7).

| Factors of Questionnaire | No of Items | Content Validity |
|---|---|---|
| | | S-CVI/Ave |
| Factor 1: knowledge on usage of plastic as food container | 7 | 0.92 |
| Factor 2: knowledge on chemical materials in plastic food container | 14 | 0.97 |
| Factor 3: knowledge on health effect of plastics usage | 16 | 0.88 |
| Factor 4: knowledge on environmental effect of plastics usage | 9 | 0.99 |
| Factor 5: knowledge on plastic categories | 14 | 0.79 |
| Factor 6: knowledge on environment related acts and regulations | 5 | 0.99 |

## 3.2. Face Validity of Questionnaire

Eleven candidates were provided with a Google Form to assess the clarity of the proposed items. Three items with an I-FVI value below 0.83 were removed: one item concerning chemicals in plastic food containers and two items from the plastic-use health effects. The S-FVI/Ave values were acceptable, as shown in Table 2. The third draft comprised 58 items with six factors that were intended for pretesting.

**Table 2.** Face validity index in cognitive debriefing phase (n = 11).

| Factors of Questionnaire | No of Items | Face Validity |
|---|---|---|
| | | S-FVI/Ave |
| Factor 1: knowledge on usage of plastic as food container | 7 | 0.96 |
| Factor 2: knowledge on chemical materials in plastic food container | 14 | 0.89 |
| Factor 3: knowledge on health effect of plastics usage | 13 | 0.89 |
| Factor 4: knowledge on environmental effect of plastics usage | 9 | 0.99 |
| Factor 5: knowledge on plastic categories | 13 | 0.87 |
| Factor 6: knowledge on environment related acts and regulations | 5 | 0.96 |

## 3.3. Pretesting Process of Validity

The third questionnaire draft was given to 30 selected food hawkers in Pasir Puteh using Google forms. The mean (SD) duration of respondents answering the third draft was 6.4 (4.9) minutes. The instructions that were given in the questionnaire were easily understood by 56.7% (17) of respondents, while 86.7% (26) commented that sentences and questionnaire arrangements were well understood. Four respondents (16.6%) took some time to understand the answer choices, and 83.3% (25) commented that the writing type and font size met the formatting requirement. Twenty respondents (66.7%) commented that the question arrangements and images were well arranged. The comments concerning the pretesting process are provided in Table 3.

**Table 3.** Comments on questionnaire in pre-testing session (n = 30).

| Comments | n (%) |
|---|---|
| Instructions given in this questionnaire: | |
| Easily understood | 17 (56.7) |
| Take some time to understand | 13 (43.3) |
| Difficult to understand on certain sentences | 0 (0.0) |
| Sentences used and arrangements in this questionnaire: | |
| Easily understood | 26 (86.7) |
| Take some time to understand | 4 (13.3) |
| Difficult to understand on certain sentences | 0 (0.0) |
| Answer choices given in this questionnaire: | |
| asily understood | 26 (86.7) |
| Take some time to understand | 4 (13.3) |
| Difficult to understand on certain sentences | 0 (0.0) |
| Type of writing and font size: | |
| Met the formatting | 25 (83.3) |
| Easy to understand | 9 (30%) |
| Need to increase the font size | 1 (3.3) |
| The arrangement of questions and images in this questionnaire: | |
| Well arranged | 20 (66.7) |
| Neat and nice | 9 (30.0) |
| Need improvement | 1 (3.3) |

### 3.4. Sociodemographic Characteristics of the Participants

A total of 660 night-market food hawkers around the Kota Bharu district responded to this questionnaire. Both genders were represented well: 50.8% (335) were female and 49.2% (325) were male. The mean age was 36.6 (12.2) years. The majority (48.9%) of the respondents had a secondary school or lower educational background, followed by a diploma or equivalent (34.2%). Most hawkers (64.2%) had three or more years of business experience; social media (63.5%) was considered the most crucial information medium on plastic use in the food business. Participants' sociodemographic characteristics are shown in Table 4.

**Table 4.** Sociodemographic background of the participants who responded in the reliability analysis questionnaire (n = 660).

| Variables | Mean (SD) | n (%) |
|---|---|---|
| Age | 36.6 (12.2) | |
| Gender | | |
| Female | | 335 (50.8) |
| Male | | 325 (49.2) |
| Educational background | | |
| No formal education | | 50 (7.7) |
| Secondary school or lower | | 323 (48.9) |
| Diploma or equivalent | | 226 (34.2) |
| Degree or equivalent | | 57 (8.6) |
| Master or equivalent or higher | | 4 (0.6) |
| Business experience | | |
| Less than 3 years | | 236 (35.8) |
| More than 3 years | | 424 (64.2) |
| Information source | | |
| Social media | | 420 (63.5) |
| Television | | 366 (55.4) |
| Newspaper | | 269 (40.7) |
| Official source | | 194 (29.3) |
| Advertisement | | 168 (25.4) |
| Radio | | 174 (26.3) |

### 3.5. Knowledge of Food Hawkers on Plastic Food Containers Using IRT Analysis

An IRT using a 'mirt' analysis showed that the output that was given by the test between a difficulty index (b) of −3 to +3 was 86.7% and all items had an acceptable discrimination index (a) value. Eleven items (KG3, KC6.b, KH1.b, KH1.c, KH2.b, KH2.d, KE1, KE2, KE4, KE7, and KR1) did not fit the model (*p*-value < 0.05) based on the goodness-of-fit test. Although these items did not fit the model, after considering each item's importance with its difficulty and discriminability index, the items were kept in the model. The model had shown to significantly fulfil all the fit indexes. The fit indexes were RMSEA = 0.087; CFI = 0.864; and TLI = 0.849 with the *p*-value of < 0.001. The AIC (30,413.17 vs. 9675.81) and BIC (30,943.26 vs. 9873.47) value also improved in the primary to the final model. The marginal reliability value was 0.77 which indicated the acceptable overall consistency of the test score in measuring the knowledge trait. From the total 58 items that were analysed using IRT, 36 items were removed, and the final model was left with 22 items. The result of item analysis by the 2-PL IRT model is summarized in Table 5.

**Table 5.** The 2-PL IRT parameter estimates and item fit statistics for knowledge score questionnaire.

| Items | Difficulty (b) | Discrimination (a) | $X^2$(df = 8) | *p*-Values |
|---|---|---|---|---|
| KG1 | −1.44 | 1.94 | 10.03 | 0.613 |
| KG2 | −3.06 | 1.10 | 10.77 | 0.210 |
| KG3 | −3.48 | 0.80 | 63.79 | **<0.001** |
| KC1 | −1.75 | 1.51 | 21.33 | 0.067 |
| KC4 | −1.63 | 1.42 | 20.10 | 0.073 |
| KC6.a | −1.50 | 2.09 | 11.90 | 0.371 |
| KC6.b | −0.63 | 2.53 | 22.30 | **0.014** |
| KC6.c | −0.19 | 2.23 | 16.63 | 0.055 |
| KH1.a | −2.85 | 1.38 | 16.32 | 0.177 |
| KH1.b | −1.55 | 1.61 | 28.33 | **0.008** |
| KH1.c | −0.98 | 1.05 | 22.61 | **0.047** |
| KH2.a | −1.47 | 2.01 | 10.14 | 0.604 |
| KH2.b | −0.48 | 1.59 | 63.65 | **<0.001** |
| KH2.d | −0.14 | 1.40 | 45.59 | **<0.001** |
| KE1 | −2.08 | 2.14 | 22.43 | **0.008** |
| KE2 | −2.24 | 1.86 | 21.07 | **0.033** |
| KE4 | −1.64 | 1.68 | 67.10 | **<0.001** |
| KE7 | −1.88 | 2.10 | 23.01 | **0.018** |
| KE8 | −1.67 | 2.36 | 16.72 | 0.116 |
| KE9 | −2.52 | 1.96 | 15.40 | 0.081 |
| KR1 | −0.08 | 1.23 | 45.87 | **<0.001** |
| KL3.a | −2.67 | 1.43 | 5.54 | 0.938 |

RMSEA = 0.087, SRMR = 0.086, CFI = 0.86, TLI = 0.85 *p*-value = < 0.001. Abbreviations: S-$X^2$ = Standardized $X^2$, RMSEA = Root Mean Square Error of Approximation, TLI = Tucker–Lewis Index, CFI = Comparative Fit Index. Items with *p*-values < 0.05 in the assessment of the item fit are highlighted in bold.

### 3.6. Final Version of Validated Knowledge Questionnaire

The final version of the validated questionnaire consisted of 22 items. Table 6 summarizes all the items with the response options of "Yes", "No" and "Don't know.

**Table 6.** Final items in questionnaire to assess knowledge level of street food hawkers to support single-use plastic reduction program in Kelantan.

| Item No. | Item |
|---|---|
| KG1 | Disposed single-use plastics will undergo chemical decomposition |
| KG2 | Plastic food containers are single-use plastics |
| KG3 | There is a campaign on "No Plastic Bag Day" in Malaysia |
| KC1 | Polystyrene is a type of plastic |
| KC4 | Using plastic containers to wrap hot food causing material from plastic to enter the food |
| KC6.a | The effects of plastic chemicals on packed foods are influenced by:(a) food temperature |
| KC6.b | (b) Duration of food stored in plastic containers |
| KC6.c | (c) Size of plastic food containers used |
| KH1.a | Plastic chemicals in food cause long-term effects to the health of consumers in these categories:(a) Baby |
| KH1.b | (b) Children |
| KH1.c | (c) Teenagers |
| KH2.a | Plastic chemicals in food packaging can increase the risk of:(a) Cancer |
| KH2.b | (b) Miscarriage |
| KH2.d | (c) Foetal malformations |
| KE1 | Reducing the use of plastics can help the government reduce the cost of environmental pollution control |
| KE2 | Plastic food packaging chemical wastes cause land pollution |
| KE4 | There are plastic food wrappers that are not decomposed by the environment |
| KE7 | Plastic waste can cause marine pollution |
| KE8 | Plastic waste can endanger marine life |
| KE9 | Plastic waste burning will cause toxic emissions to the environment |
| KR1 | Plastics are classified into 7 groups based on the materials used in its production |
| KL3.a | There are laws that prohibit ready-to-eat food from being wrapped with newspaper |

## 4. Discussion

The main aim of this study is to develop and validate a new culturally acceptable questionnaire to assess the knowledge of street food hawkers, in order to support the single-use plastics reduction program in Kelantan, Malaysia. This questionnaire was built in Malay language to make it culturally acceptable.

Generally, the items in this psychometric tool were relevant, understandable, and discriminating between overall best and overall worst candidate as evidenced by the difficulty parameters between −3 and +3 (b) and discrimination parameters (a). An IRT analysis showed a good amount of information on knowledge given by the test between a −3 and +3 ability range from the final 22 items. In our analysis, 86.7% of knowledge information could be provided on the assessment of knowledge on single-use plastic by the food hawkers that were studied. Moreover, questions that were too hard or too easy for examinees would provide little information about their abilities and so they were removed in our analysis. However, two items exceeded the difficulty index cut-off point value, which were KG2 (b = −3.06) and item KG3 (b = −3.48). Both items were kept, as their information was important in the assessment of the general knowledge of single-plastic use among hawkers. In this study, the item showed a good discrimination index between 0.8 and 2.36. The marginal reliability was 0.77, showing that this questionnaire is a reliable

tool to be used to estimate the knowledge of street food hawkers, in order to support the single-use plastic reduction program.

This study demonstrated that the final version of the knowledge domain showed good psychometric properties with relevant content, and a good response process with internal validity evidence. The initial 58 knowledge items in this questionnaire were proposed in six domains but underwent tremendous item removal, involving a total of 32 items removed and 22 items remaining. All the items were grouped in a single domain with the aim of producing a comprehensive, clearly defined questionnaire, non-redundant items, and a non-exhaustive tool. IRT can be used to evaluate the psychometric properties of an existing scale and its items, to optimally shorten the scale when necessary, and to evaluate the performance of the reduced scale. When used appropriately, IRT modelling can produce precise, valid, and relatively brief instruments, resulting in minimal response burden [38]. However, the psychometric properties in this study could not be compared to previous study on the knowledge domain on the single-use plastic food container because of inadequate information and the different method that was used for the knowledge assessment [17,18,44].

There are several strengths of this study. First, to the best knowledge of the researchers' team, this study is one of the first published studies on the development and validation of the knowledge of street food hawkers, in order to support the single-use plastics reduction program in Kelantan and generally, in Malaysia. This process was culturally adapted and greater attention was given to food hawkers' use of single-use plastic food containers in their business. Hence, this validated tool will be useful to assess the knowledge of other food hawkers around the country, in order to support the single-use plastics program in Malaysia. This can contribute to the reduction in environmental pollution from the excessive use of single-use plastics.

There are also limitations to this study, as it was confined to Kelantan state, representing only the north-eastern part of Malaysia. Cross validation of this study is suggested to involve other races and ethnicities in other part of Malaysia to improve the validity and reliability of this questionnaire. Moreover, the data collection was restricted due to the COVID-19 pandemic, and the use of single-use plastic food containers surged during this pandemic period.

## 5. Conclusions

In this study, a new Malay-validated questionnaire was developed and validated among a sample of street food hawkers in Kelantan, located in a north-eastern state in Peninsular Malaysia. The questionnaire consisted of 22 items on the knowledge of street food hawkers, in order to support the single-use plastics reduction program in Kelantan. The items in the knowledge domain were psychometrically valid and based on an IRT analysis, with a good amount of information included for the −3 to +3 ability difficulty range. A total of 36 items were removed from the domain's initial 58 items. The newly developed questionnaire showed good psychometric validity according to the IRT analysis. The IRT analysis was a valid and reliable tool to assess the knowledge of food hawkers, in order to support the single-use plastics reduction program, incorporating knowledge on food hawkers' awareness of the plastic reduction program, the chemicals that are used in single-use plastic food containers, and the health and environmental impact of excessive usage of single-use plastic food containers.

**Author Contributions:** N.B.A.A.: Conceptualization, methodology, validation, formal analysis, investigation, visualization, writing—original draft, funding acquisition. A.F.I.: Conceptualization, methodology, writing—review and editing, supervision, resources. N.A.Y.: Conceptualization, methodology, writing—review and editing, resources. All authors have read and agreed to the published version of the manuscript.

**Funding:** This research is funded by Tabung Insentif Pembangunan Pengajian Siswazah (TIPPS), Universiti Sains Malaysia (USM).

**Institutional Review Board Statement:** Ethical approval was obtained from the Human Research Ethics Committee, Universiti Sains Malaysia (USM)—USM/JEPeM/19110814.

**Informed Consent Statement:** All the participants were given an informed consent to be signed upon they consented to participate in this study.

**Data Availability Statement:** The data presented in this study are available on request from the corresponding author. The data are not publicly available due to privacy considerations.

**Acknowledgments:** The authors would like to show gratitude to all the street food hawkers who participated in the study. We would also like to dedicate our special thanks to the Local Council Division of the Kelantan State Secretary Office for the great collaboration.

**Conflicts of Interest:** All authors declare no competing interest in this study.

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
