# Peer review of "Development and Validation of a New Questionnaire to Measure Knowledge Level of Street Food Hawkers to Support the Single-Use Plastics Reduction Program in Kelantan, Malaysia"

_sustainability, doi:10.3390/su14137552_

Round 1
Reviewer 1 Report
Comments to Authors: sustainability-1597771
Abstract
- Add information about experts/selected representatives that had important role to develop the questionnaire.
- In lines 12-13, it was mentioned that the food hawkers filled out a questionnaire containing 22 items, while when I read thoroughly in the methodology and result, the last version of the questionnaire (3rddraft) is contained 59 items that later evaluated through IRT and yielded that 22 items are fit and validated as the result of this study (Table 6 and 7). It raised a question in my mind, who did judge the 59 items questionnaire? (Provide the clearer explanation in methodology).
Background
- Line 46-47. Please confirm again with other sources about the consumption of single-use plastics per capita per day in Malaysia. In my opinion, the number that you cited is too high.
- Line 74. Previous studies “only” stressed… It is better to use the word "mostly" to avoid a wrong judgment, or because there is still not enough exploration. In the next line, it is necessary to emphasize the few research on food hawkers.
- The purpose of this paper is to develop and validate a questionnaire to assess the knowledge level on single-use plastic reduction program. However, the fundamental reason of the importance of the validation of questionnaire is not available in the background. You can also argue what happen if the questionnaire is not validated. Please elaborate this part.
Methodology
- What is the reason to use a purposive sampling method to get 660 respondents (food hawkers)?
- Line 203. Is it ≤ – 3.0 or ≥ 3.0 ? Please confirm it.
- Connected to the question in the abstract, provide the clearer step-by-step questionnaire development, who did take part for the judgement of each questionnaire draft?.
Result
- Sentences in Lines 204-206 have been available in methodology. It is better to omit or elaborate the important point in the methodology.
- Sentences in Lines 228-230 are repetition. It is better to be shortened and placed in methodology section (2.3.1)
- Line 240. The retained items should be 58 (61-3). Please confirm it.
- Table 6-7. If it possible, just provide the English version in order to reduce the length.
Discussion
- Line 311, I think there is a missing word. Unclear sentence.
- Line 310-313, it needs more discussion about the retained and removed items from the list in this study, why they were removed. Based on your literature exploration and comparison to the study result, do you think that the 22 retained items have been adequate to evaluate the public knowledge on the single use plastics? Compare the result (validity test) of this study to other social survey methods (such as the reliability test using Cronbach’s alpha, or factor analysis to retain or remove items, or any other methods). Cite some related papers to explain or/and to compare the result.
- It needs to provide broader implication of the study to a social science. Such as, how important is step-by step method to develop questionnaire and the validation method in order to improve the quality of questionnaires or to provide the appropriate questionnaires as the most used tool in the social research method.
Conclusion
- Elaborate brief information about 22 items of the questionnaire to assess the knowledge levels of food hawkers on the single-use plastic food containers based on your finding.
Author Response
Abstract
- Add information about experts/selected representatives that had important role to develop the questionnaire. Response: The details were discussed in the methodology part.
- In lines 12-13, it was mentioned that the food hawkers filled out a questionnaire containing 22 items, while when I read thoroughly in the methodology and result, the last version of the questionnaire (3rddraft) is contained 59 items that later evaluated through IRT and yielded those 22 items are fit and validated as the result of this study (Table 6 and 7). It raised a question in my mind, who did judge the 59 items questionnaire? (Provide the clearer explanation in methodology). Response: The issues have been reviewed with explanation in methodology, including tables 5 and 6)
Background
- Line 46-47. Please confirm again with other sources about the consumption of single-use plastics per capita per day in Malaysia. In my opinion, the number that you cited is too high. Response: Line 46-47 has been reviewed with a new citation. “In a 2019 study commissioned by WWF, Malaysia has the highest annual per capita plastic use, at 16.78 kg per person compared to China, Indonesia, Philippines, Thailand and Vietnam. In terms of plastic waste, Malaysia ranks the second highest in overall generated waste”
- Line 74. Previous studies “only” stressed… It is better to use the word "mostly" to avoid a wrong judgment, or because there is still not enough exploration. In the next line, it is necessary to emphasize the few research on food hawkers. Response: Few studies are provided.
The word “only “has been changed to “mostly” (line 64). - The purpose of this paper is to develop and validate a questionnaire to assess the knowledge level on single-use plastic reduction program. However, the fundamental reason of the importance of the validation of questionnaire is not available in the background. You can also argue what happen if the questionnaire is not validated. Please elaborate this part. Response: The fundamental reason for questionnaire development and validation and its importance is explained further in the text.
Methodology
- What is the reason to use a purposive sampling method to get 660 respondents (food hawkers)? Response: The purposive sampling method was done to purposely to get the suitable respondents, with the restriction due to the COVID-19 pandemic
- Line 203. Is it ≤ – 3.0 or ≥ 3.0? Please confirm it. Response: Difficulty parameters had been reviewed in line 214 “Items with difficulty parameter of <3.0 or >3.0 will be discarded”
3. Connected to the question in the abstract, provide the clearer step-by-step questionnaire development, who did take part for the judgement of each questionnaire draft? Response: The drafted questionnaires were vetted by experts in the field consisting of environmental health specialists, local authority representatives and hawkers’ association representatives.
a. Questionnaire development (1st draft) involved 6 representatives and is explained in lines 116-145.
b. Content validation (2nd draft) involved 4 expert panelists listed in lines 163 – 166.
c. Face validity (3rd draft) involved 11 respondents listed in lines 177-179.
d. Pretesting (3rd draft) involved 30 respondents as mentioned in lines 185-187.
f. Item response theory (IRT) (4th draft) as mentioned in lines199 - 207
Result
- Sentences in Lines 224-226 have been available in methodology. It is better to omit or elaborate the important point in the methodology. Response: Line 224 -226 were omitted. The result in section 3.1 is reviewed in line 255-256.
- Sentences in Lines 228-230 are repetition. It is better to be shortened and placed in methodology section (2.3.1) Response: Lines 228 -230 haves been deleted. And replaced with lines 262-263.
- Line 240. The retained items should be 58 (61-3). Please confirm it. Response: The retained items were 58 instead of 59. Corrected in line 269.
- Table 6-7. If it possible, just provide the English version in order to reduce the length. Response: Malay version of items had been deleted as in Table 6. All items have been listed in Table 6. Table 7 had been deleted.
Discussion
- Line 311, I think there is a missing word. Unclear sentence. Response: “ the missing word is domains” The initial 58 knowledge items in this questionnaire were proposed in 6 domains but had undergone tremendous item removal, where it involved total 37 items removal and left with 22 items as in line 351
- Line 310-313, it needs more discussion about the retained and removed items from the list in this study, why they were removed. Based on your literature exploration and comparison to the study result, do you think that the 22 retained items have been adequate to evaluate the public knowledge on the single use plastics? Compare the result (validity test) of this study to other social survey methods (such as the reliability test using Cronbach’s alpha, or factor analysis to retain or remove items, or any other methods). Response: Retained and removed items criteria are discussed lines 334 -347. The amount of information provided by the 22 items on knowledge is 86.7% which is good. (line 336-340).
- . Cite some related papers to explain or/and to compare the result. Response: There is inadequate information on knowledge assessment using IRT analysis among food hawkers as in lines 358-359
- It needs to provide broader implication of the study to a social science. Such as, how important is step-by step method to develop questionnaire and the validation method in order to improve the quality of questionnaires or to provide the appropriate questionnaires as the most used tool in the social research method. Response: As in lines 354- 358
Conclusion
- Elaborate brief information about 22 items of the questionnaire to assess the knowledge levels of food hawkers on the single-use plastic food containers based on your finding. Response: Conclusion had been revised in lines 377-385
Reviewer 2 Report
The authors created a special questionnaire to measure knowledge level of Malaysian street food hawkers. The idea for creating this questionnaire is that individuals with more knowledge of environmental issues are likely to demonstrate pro-environmental behavior.
The manuscript is well written and clearly structured. The statistical analyses were conducted properly. I have only several comments.
Introduction:
I recommend a little bit expand introduction, namely in terms of effects of knowledge on pro-environmental behavior (only one paper is referred), as well as quote more research concerning knowledges and attitudes of food handlers (if any?).
Procedure and Results
The authors describe very carefully and in details the several stages of the process of creating the questionnaire. But the tables should be restructured.
Table 1, Table 2 It is not clear, why only the one item „knowledge“ is stated in the first column. I think it is superfluous. The second column should be right-aligned.
Table 3, Table 4 The first column should be right-aligned.
Table 6, Table 7 The second column should be right-aligned.
Table 6, Table 7 Because the paper is in English, I would be better to present the questionnaire items in the tables in English and the original items in Malay language move to an attachment.
References should be in the format recommended by the Sustainability journal.
Author Response
Introduction:
I recommend a little bit expand introduction, namely in terms of effects of knowledge on pro-environmental behavior (only one paper is referred), as well as quote more research concerning knowledges and attitudes of food handlers (if any?). Response: Some information added in lines 55 - 59
Procedure and Results
The authors describe very carefully and in details the several stages of the process of creating the questionnaire. But the tables should be restructured.
Table 1, Table 2 It is not clear, why only the one item „knowledge“ is stated in the first column. I think it is superfluous. The second column should be right-aligned.
Table 3, Table 4 The first column should be right-aligned.
Table 6, Table 7 The second column should be right-aligned.
Table 6, Table 7 Because the paper is in English, I would be better to present the questionnaire items in the tables in English and the original items in Malay language move to an attachment.
Response:
Table 1 explained the CVI analysis result in validation phase.
Table 2 explained the FVI analysis result in validation phase
-word” knowledge” is the knowledge domain that consist of initial 65 items in 6 factors”
-Table 3 , Table 4 had been justified
- Malay version of items had been removed
- Table 6 and Table had been combined and aligned
References should be in the format recommended by the Sustainability journal.
Response:
Reference format uses as in American Chemical Society
Reviewer 3 Report
Reviews of
Development and Validation of a New Questionnaire to Measure Knowledge Level of Street Food Hawkers to Support the Single-use Plastics Reduction Program in Kelantan
Summary of the Manuscript
The focus of the manuscript is to assess the knowledge level of street hawkers on single-use plastic food containers.
Overall, the theoretical arguments posed in this manuscript is praiseworthy, however, the manuscript needs to go through further reviews. Specially, in some segments the manuscript needs to discuss more clearly about analytical procedure and theoretical arguments.
Manuscript Comments:
It is certain that authors have attempted to explain complex components of assessing knowledge level of street hawkers on plastic container use and attempted to develop a scale with appropriate psychometric properties. For better clarity, author should consider following general and specific comments.
- Authors are suggested to drop CVI results in abstract section and discuss these results from general point of view, and for general audiences.
- [Study design and participant] It is strongly suggested that authors should incorporate more specific demographic information about Kelantan and Kota Bharu area. The information provided is not sufficient to portray the density of the population or overall living condition.
- [184] Please check the citation format for Yusof (2019).
- [190] Authors are suggested to check the statement. It is ambiguous.
- [195] Authors are also suggested to provide some references and logics behind selecting IRT models for the scale validation. The process is relatively new, and rarely used. Readers should understand the reasons behind using IRT. Author are suggested to extend this section.
- In the methodology section, authors are suggested to discuss the IRT assumptions and their study conditions.
- [208-211, 215] Authors are suggested to rephrase these sentences.
- [219] Please check the reference.
- [222] Authors my think of merging the development of questionnaire section in 2.2 for better understanding and start the results section from 3.2.
- [274] Authors are suggested to explain BIC and AIC values for reference model and final model. This information is ambiguous.
- [Table 7] Its difficult to understand the table. Authors are suggested to reformat the table. Also, it will be easier to read if the answers are listed separately.
- [results] Authors are suggested to provide references to support CFI, TLI values presented in the study.
- Overall, authors have done a great job developing the scale.
Author Response
Summary of the Manuscript
The focus of the manuscript is to assess the knowledge level of street hawkers on single-use plastic food containers.
Overall, the theoretical arguments posed in this manuscript is praiseworthy, however, the manuscript needs to go through further reviews. Specially, in some segments the manuscript needs to discuss more clearly about analytical procedure and theoretical arguments.
Response: Has been explained in lines 59 - 63.
- Authors are suggested to drop CVI results in abstract section and discuss these results from general point of view, and for general audiences. Response: the abstract has been reviewed by removing CVI and FVI results
- [Study design and participant] It is strongly suggested that authors should incorporate more specific demographic information about Kelantan and Kota Bharu area. The information provided is not sufficient to portray the density of the population or overall living condition. Response: Demographic information of Kelantan is provided in lines 105-112.
- [184] Please check the citation format for Yusof (2019). Response: Redone as "For objective assessment of content validity, item-level content validity index (I-CVI) , scale-level content validity index (S-CVI) and S-CVI/Ave based on formula by Yusof [32]."
- [190] Authors are suggested to check the statement. It is ambiguous. Response: The statement is reviewed in lines 227-229
- [195] Authors are also suggested to provide some references and logics behind selecting IRT models for the scale validation. The process is relatively new, and rarely used. Readers should understand the reasons behind using IRT. Author are suggested to extend this section. Response: The IRT information is provided in lines 207-212, and 235- 245
- In the methodology section, authors are suggested to discuss the IRT assumptions and their study conditions. Response: Explained in lines 235 - 243
- [208-211, 215] Authors are suggested to rephrase these sentences. Response: sentence in line 208-211, 255 had been rephrased in line 246-250
- [219] Please check the reference. Response: Revised reference in line 219 done (lines 259-260)
- [222] Authors my think of merging the development of questionnaire section in 2.2 for better understanding and start the results section from 3.2. Response: Reviewed in line 264
- [274] Authors are suggested to explain BIC and AIC values for reference model and final model. This information is ambiguous. Response: Reviewed done in lines 252 - 258
- [Table 7] Its difficult to understand the table. Authors are suggested to reformat the table. Also, it will be easier to read if the answers are listed separately. Response: Table 7 has been removed and combined with table 6.
- [results] Authors are suggested to provide references to support CFI, TLI values presented in the study. Response: Reference used is [38]: Dimitrov, D. M. Marginal True-Score Measures and Reliability for Binary Items as a Function of Their IRT Parameters: Appl. Psychol. Meas. 2016, 27 (6), 440–458. https://doi.org/10.1177/0146621603258786.
- Overall, authors have done a great job developing the scale. Response: Thank you for your kind consideration.
Round 2
Reviewer 1 Report
Comments and Suggestions for Authors
Dear authors,
Thank you for taking the time to revise this manuscript. I think it has greatly improved the clarity of your messages. There are some areas where I feel some modifications are still needed. I hope the comments are useful.
Line 56
Briefly add other respondents’ roles (experts, researcher, local authorities, public health specialist, toxicologist, lecturer/linguistic) in developing questioners as it is presented in your analysis and result (CVI, FVI). Because they are the part of the process before the last draft assessed by the 660 street food hawkers.
line 363
------between June 2020 and February 2021. (…………). Same comment in the abstract
You need briefly add other respondents (experts, local authorities, ect.) in developing your questionnaires.
After Line 585
You can add a framework of your step-by step drafting questioner and analysis process (validation) to make it easier understand by readers.
Line 689: better remove “hypnotized” because in this article (Manuscript) you do not develop hypothesis/ hypotheses.
in L706
It is mentioned about the 6 domains, however in Table 1, it is only available 1 domain (knowledge). Instead, I saw "factor" in Table 1. It should be consistent to use “domain” or “factor”. (The same comment to Table 2)
After Line 1159. (end of the conclusion).
You can add a final sentence to conclude your finding (For example):
To assess the knowledge of street food hawkers to on plastic waste, the items such as the attention to single used plastic and polystyrene, characteristic of plastics, level of education (campaign) to hawkers, the effects and the influencing factors of plastic to human body and the environment, and how to reduce the plastic use should be include in questioners.
Author Response
1. Line 56
Briefly add other respondents’ roles (experts, researcher, local authorities, public health specialist, toxicologist, lecturer/linguistic) in developing questioners as it is presented in your analysis and result (CVI, FVI). Because they are the part of the process before the last draft assessed by the 660 street food hawkers.
Amendments made are:
in line 170-185.
2. line 363
------between June 2020 and February 2021. (…………). Same comment in the abstract
You need briefly add other respondents (experts, local authorities, ect.) in developing your questionnaires.
Amendments made are:
Line 12-15 in abstract and line 171-172.
3. After Line 585
You can add a framework of your step-by step drafting questioner and analysis process (validation) to make it easier understand by readers.
Amendments made are:
Figure 1 is attached to summarized the whole questionnaire development and validation process.
4. Line 689: better remove “hypnotized” because in this article (Manuscript) you do not develop hypothesis/ hypotheses.
Amendments made are:
line 262-263. the hypothesized model is replaced with the initial proposed model.
5. in L706
It is mentioned about the 6 domains, however in Table 1, it is only available 1 domain (knowledge). Instead, I saw "factor" in Table 1. It should be consistent to use “domain” or “factor”. (The same comment to Table 2)
Amendments made are:
both domain column in Table 1 and Table 2 are deleted.
6. After Line 1159. (end of the conclusion).
You can add a final sentence to conclude your finding (For example):
To assess the knowledge of street food hawkers to on plastic waste, the items such as the attention to single used plastic and polystyrene, characteristic of plastics, level of education (campaign) to hawkers, the effects and the influencing factors of plastic to human body and the environment, and how to reduce the plastic use should be include in questioners.
Amendments made are:
Line 409-414.